# APOLLO 🚀 : Unified Adapter and Prompt Learning for Vision Language Models

**Sanjoy Chowdhury**[1*]    **Sayan Nag**[2*]    **Dinesh Manocha**[1]

[1]University of Maryland, College Park    [2]University of Toronto

{sanjoyc,dmanocha}@umd.edu    sayan.nag@mail.utoronto.ca

Project page – https://gamma.umd.edu/pro/vision_language/apollo/

## Abstract

The choice of input text prompt plays a critical role in the performance of Vision-Language Pretrained (VLP) models such as CLIP. We present APOLLO , a **unified multi-modal approach** that combines Adapter and Prompt learning for Vision-Language models. Our method is designed to substantially improve the generalization capabilities of VLP models when they are fine-tuned in a few-shot setting. We introduce trainable cross-attention-based adapter layers in conjunction with vision and language encoders to strengthen the alignment between the two modalities. We enforce consistency between the respective encoder branches (receiving augmented inputs) to prevent overfitting in downstream tasks. Our method is evaluated on three representative tasks: generalization to novel classes, cross-dataset evaluation, and unseen domain shifts. In practice, APOLLO achieves a **relative gain up to 6.03%** over MaPLe (SOTA) on novel classes for 10 diverse image recognition datasets.

## 1 Introduction

Recent years have witnessed tremendous success of largescale pre-trained language models (Devlin et al., 2018; Liu et al., 2019; Raffel et al., 2020; Biderman et al., 2023; Scao et al., 2022; Shen et al., 2023; Touvron et al., 2023) as well as visual models (Dosovitskiy et al., 2020; Liu et al., 2021b; Tan and Le, 2019) leading to a surge in pre-trained Vision-Language models (Dou et al., 2022; Li et al., 2022a; Yang et al., 2022; Wang et al., 2023a; Li et al., 2023b,a; Wang et al., 2023b) for multi-modal downstream tasks. Despite being largely successful in terms of generalization capabilities, these VLP models such as CLIP (Radford et al., 2021) are difficult to fine-tune for few-shot learning-based downstream tasks (Khattak et al., 2023). This is mainly because of the massive scale of these models coupled with the deficiency of training data

---

*Equal contribution.

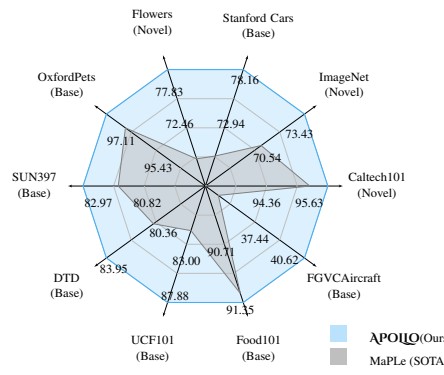

Figure 1: APOLLO **sets a new state-of-the-art** across several datasets for the base-to-novel class generalization task by learning a vision-language representation using a unified adapter and prompt tuning strategy under a few-shot setting. Each axis denotes an accuracy value, either on base or novel class (specified accordingly) for the corresponding dataset (refer to Table 1).

(Khattak et al., 2023). Recent studies in literature propose fine-tuning of these models by introducing and calibrating the parameters by prompting (Zhou et al., 2022a) and adapter (Gao et al., 2021a) techniques. While the former injects tunable parameters through learned embeddings in one or more modalities, the latter incorporates in-situ adjustments conventionally near the prediction head.

The effectiveness of such fine-tuning-based methods notwithstanding recent investigations reveal (Zhou et al., 2022b) the caveats of these approaches – such as neglecting the useful knowledge gained from the preceding pre-training step and overfitting with the downstream tasks. Although we find a considerable wealth of studies that involve text-based prompt learning (Lu et al., 2022; Shu et al., 2022; Huang et al., 2022), the same for the visual pipeline still remains an under-explored area.

**Main Results:** We present APOLLO , a novel unified adapter and prompt learning approach for VLP models to tackle the generalizability problems in few-shot scenarios. We enforce consistency between intra-modal encoders using consistency-

guided contrastive loss (Wei et al., 2020). This is done to teach the association between the query and the semantically similar negative in-batch keys. To further enhance cross-modal alignment, we employ cross-attention in the modality-specific adapter layers. This leads to better awareness of the multi-modal features. Experimental results over a varied range of recognition datasets demonstrate the efficacy of our approach. Some novel aspects of our work include:

**(1)** To the best of our knowledge, ours is the **first method that combines adapter and prompt tuning for VLP models (e.g., CLIP) in a unified manner**. This facilitates learning new tasks in a few-shot setting without compromising on their zero-shot generalizability.

**(2)** We propose a **novel multi-modal augmentation strategy** by leveraging LLMs to generate descriptive texts as augmented samples in the text branch, and text-conditioned diffusion models to generate image augmentations for the image branch.

**(3)** Our **novel application of multi-modal cross-attention adapter layers** bridges the gap between the two modalities by generating text-guided visual features and vice-versa. This promotes the synergy between the two modalities.

**(4)** Extensive evaluation on 10 challenging datasets demonstrates the effectiveness of APOLLO as **it outperforms existing methods by a significant margin** and set a new SOTA (Figure 1) for a range of downstream tasks including base-to-novel generalization, cross-dataset recognition, and domain generalization.

## 2  Related Works

### 2.1  Vision Language Models

Recent research has indicated that effectively mining image-text pairs can enable VLP models to achieve highly satisfactory results on relevant downstream tasks when compared against uni-modal frameworks. For example, models like CLIP (Radford et al., 2021), ALIGN (Jia et al., 2021), and Florence (Yuan et al., 2021) demonstrate exceptional performance on a wide range of few-shot and zero-shot visual recognition tasks. However, they are impractical to adapt to challenging downstream tasks. Prior work include specially designed approaches pursuing object detection (Bangalath et al., 2022; Zang et al., 2022; Zhou et al., 2022c) and, few-shot image recognition (Zhang et al.,

2021; Gao et al., 2021b; Kim et al., 2021) that fare much better compared to off-the-shelf VLP models (Radford et al., 2021; Jia et al., 2021). In this paper, we present a novel image and text augmentation-based technique for a unified adapter and prompt learning to enable CLIP based model to generalize well under few-shot and zero-shot visual recognition tasks.

### 2.2  Prompt Tuning

Prompt tuning (Li and Liang, 2021; Liu et al., 2021a; Lester et al., 2021) typically refers to prepending language instructions to the input text to facilitate a better understanding of the downstream tasks. For example, instead of feeding the system with a fixed template 'a photo of a <CLASS>', task-specific additional information could be more helpful for the model. More so, tuning-based methods achieve comparable performance as full finetuning but with $\sim$1000× fewer parameters. To this end, Context Optimization(CoOp) (Zhou et al., 2022b) proposes a replacement of the hand-crafted prompts with the learnable soft prompts. However, their approach lacks generalizability and demonstrates suboptimal performance under a zero-shot setting. CoCoOp (Zhou et al., 2022a) generates an image-conditional context along with text conditioning through prompt tuning, while ProDA (Lu et al., 2022) incorporates the prompt's prior distribution learning. ProGrad (Zhu et al., 2022) performs prompt updates where the gradients are aligned with the original prompt's knowledge. Recent works (Bahng et al., 2022; Khattak et al., 2023) leverage a multi-modal prompting (image + text) technique to exploit the natural association across modalities.

### 2.3  Adapter Tuning

Adapter-based methods (Houlsby et al., 2019) inject additional trainable parameters into a pre-trained model to facilitate customized learning for downstream tasks. Yuan et al. (2021); Sung et al. (2022) introduce additional layers near the prediction head to enrich pre-trained models with additional parameters. Initial efforts to develop adaptive methods for computer vision tasks involved incremental learning (Rosenfeld and Tsotsos, 2018) and domain adaption methods (Rebuffi et al., 2017, 2018). APOLLO leverages adapter and prompting to boost the performance of the model for downstream tasks, while contrastive-consistency loss ensures the generalizability of the model.

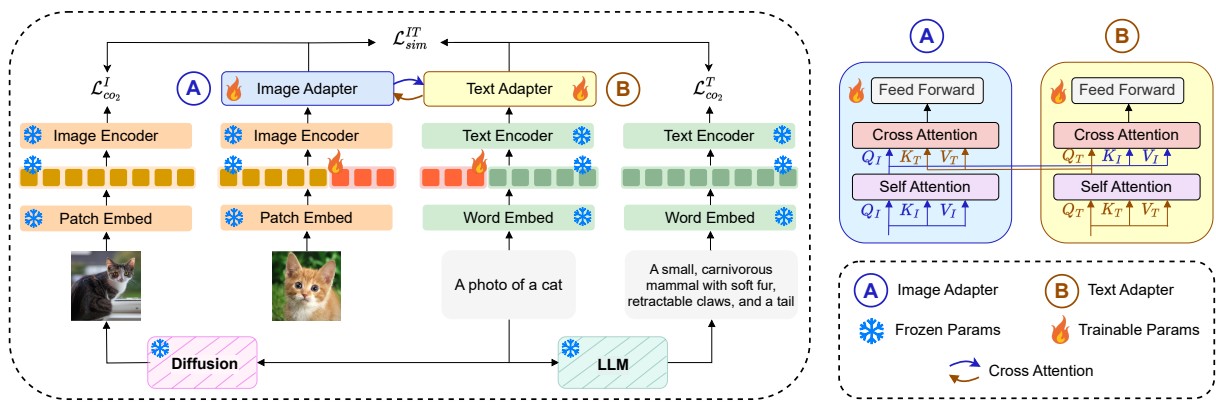

Figure 2: **Overview of the proposed** ΛPOⱢⱢO **framework** for a unified multi-modal adapter and prompt learning in VLP models. ΛPOⱢⱢO unifies prompt and adapter tuning for both the image and text branches. The image (blue) and text (yellow) adapter layers are coupled with each other through cross-modal interactions via attention which further improves the alignment between the two modalities. Each modality contains two branches receiving augmented versions of input texts and images generated using LLM and text-conditioned diffusion models respectively.

## 3 Methodology

### 3.1 Brief Review of CLIP

Among the existing Vision-Language models, CLIP (Radford et al., 2021) demonstrates strong generalizability for zero-shot image recognition. Being a multi-modal framework it's comprised of image and text encoder modules. In our set-up, we deploy a transformer-based image encoder ($\theta$) and text encoder ($\phi$).

#### 3.1.1 Image Encoder

Our patch-based image embedding module is comprised of $\mathcal{K}$ transformer layers that split an image into $\mathcal{M}$ fixed-sized patches which are projected into patch embeddings $\mathcal{E}_0 \in \mathbb{R}^{\mathcal{M} \times d_v}$. Patch embedding $\mathcal{E}_i$ is fed into the next transformer block combined with a learnable class token $c_i$ where $i \in \{1, \dots \mathcal{K}\}$. Therefore,

$$[c_{i+1}, \mathcal{E}_{i+1}] = \theta_{i+1}([c_i, \mathcal{E}_i]) \quad (1)$$

The final image representation is obtained by projecting class token $c_k$ of the last transformer layer to a common latent space:

$$u = \mathcal{I}_{Proj}(c_{\mathcal{K}}) \, ; u \in \mathbb{R}^{d_{vl}}. \quad (2)$$

#### 3.1.2 Text Encoder

Text encoder generates feature embedding by tokenizing the word sequence. At the $(i + 1)^{th}$ step $\mathcal{W}_i$ is fed as input to the transformer text encoder:

$$[\mathcal{W}_{i+1}] = \phi_{i+1}(\mathcal{W}_i) \, ; \forall i \in \{1, \dots \mathcal{K}\}. \quad (3)$$

Like its image counterpart, the final text representation $v$ is obtained by projecting the text embedding of the last transformer block $\phi_{\mathcal{K}}$ to a shared latent space:

$$v = \mathcal{T}_{Proj}(w_{\mathcal{K}}) \, ; v \in \mathbb{R}^{d_{vl}}. \quad (4)$$

#### 3.1.3 Zero-Shot Classification

During zero-shot inference, text prompts are provided with class labels $y \in \{1, 2, \dots C\}$. Class label $\hat{y}$ with the highest cosine similarity score $(\text{sim}(\cdot))$ w.r.t the input image $x$ is computed, where $\tau$ is the temperature parameter.

$$p(\hat{y} \mid x) = \frac{\exp\left(\text{sim}\left(u, v_{\hat{y}}\right)/\tau\right)}{\sum_{i=1}^{C} \exp\left(\text{sim}\left(u, v_i\right)\right)}. \quad (5)$$

This template-based approach was found to produce suboptimal results across domains. To address this, CoOp (Zhou et al., 2022b) proposed an alternative approach that replaces the hand-crafted prompt with a set of learnable context vectors that generate task-specific text embeddings. These are augmented with the previously generated word embeddings. Likewise, learnable context vectors are concatenated with the patch embeddings. We extend MaPLe (Khattak et al., 2023) to present a novel multi-modal prompt learning approach to ensure generalizability across different downstream tasks under a zero-shot classification setting.

### 3.2 Our Approach: ΛPOⱢⱢO

We note that prior methods are based on uni-modal prompt tuning approaches to finetune CLIP to perform downstream image recognition. On the other hand, MaPLe is the only work that uses multi-modal prompt tuning. In a major departure from this approach, we introduce a novel unified adapter

and prompt learning method with a novel augmentation strategy to the text and image inputs to facilitate further regularization. To this end, we supplement our adapter layers with multi-modal cross-attention modules that ensure the text and the image modalities are aligned. The text input is passed through a pre-trained LLM to obtain a more descriptive expression of itself while the image branch comprises a text-conditioned diffusion model to obtain augmentation of the input image as shown in Figure 2.

### 3.2.1 Image and Language Prompting

We facilitate deep image and language prompting across the hierarchies in the image and text transformer backbones (of CLIP) by introducing learnable tokens at every transformer layer. It has been shown (Khattak et al., 2023) that systematic prompt sharing in successive stages is more intuitive over independent prompts as consecutive transformer blocks ensure densely correlated feature learning.

### 3.2.2 Input Augmentation

Generative data augmentation schemes have been leveraged very recently (Hu et al., 2023; Trabucco et al., 2023; Azizi et al., 2023; Shipard et al., 2023; Yin et al., 2023; Whitehouse et al., 2023), however, not in a multi-modal context (Appendix A.4). In our method, we add regularization by using augmented versions of respective modality inputs in the corresponding frozen encoder branches. In particular, for the text branch, we employ a pre-trained frozen LLM (Brown et al., 2020) to generate sentences about the object referred to in the text. These sentences are typically descriptions of the usual characteristics of the object. For example, as shown in Figure 2, we provide a hand-crafted template ('a photo of a <CLASS>') as input to the LLM. Considering **Cat** as the **'CLASS'**, LLM outputs a descriptive text as: **'a small carnivorous mammal with soft fur, retractable claws, and a tail'**. In this regard, KgCoOp (Yao et al., 2023) also introduces textual augmentations. However, they get an average embedding from a pre-defined number of sentences, whereas we restrict our model to generate a single sentence (on the fly) which is easier and more diverse. For the image branch we introduce a text-conditioned diffusion (Rombach et al., 2022) based augmentation strategy that generates novel examples of the same class (see Figure 2). Experimental results back our claim that adding these two separate prompting modules indeed results in enhanced performance (also see Section 4.4 for detailed ablations).

### 3.2.3 Multi-modal Cross-attention Adapter (MCA):

Lately, adapter layers have played a pivotal role in model finetuning for adaptation to new downstream tasks (Yuan et al., 2021; Li et al., 2022b; Chen et al., 2022; Hao et al., 2023). Typically, these adapter layers are trainable parameters that are added on top of an encoder to transform the embedding vector to better adapt to new tasks. Inspired by Chen et al. (2022), we maintain trainable cross-attention-based adapter layers, for the respective encoders individually. In a major departure from MaPLe, where the authors used multi-modal prompt coupling, we introduce cross-attention modules that enforce inter-modal alignment.

### 3.2.4 Loss Formulation

Our training paradigm involves the computation of two types of losses between the multi-modal encoders.

**Intra-modal Contrastive Consistency:** In a typical contrastive learning setting, the heterogeneous similarities between the query and other in-batch **negatives** are ignored resulting in suboptimal representation learning. In order to mitigate this **'class-collision'** problem and better deal with sampling bias (Wei et al., 2020) proposed a consistency-based contrastive loss. We employ this loss to enforce consistency between the respective intra-modality branches as given below:

$$\mathcal{L}_{con} = \alpha_1 * \mathcal{L}_{CO_2}^I + \alpha_2 * \mathcal{L}_{CO_2}^T. \quad (6)$$

In practice we keep $\alpha_1 = \alpha_2 = \alpha$ for all experiments. We add more details about $\mathcal{L}_{CO_2}$ in the Appendix A.2.2.

**Inter-modal Similarity Maximization:** The inter-modality loss is responsible for maximizing the similarity between the multi-modal branches. The image-text pairs in a mini-batch are passed through the respective encoders followed by the adapter layers. The normalized image and text embeddings are subsequently utilized to compute the inter-modality Image-Text Contrastive (ITC) loss given as $\mathcal{L}_{sim}^{IT}$ (refer to Appendix A.3 for further details).

The total loss is calculated by the weighted average of the above two losses:

$$\mathcal{L}_{total} = \mathcal{L}_{sim}^{IT} + \alpha * \mathcal{L}_{con}. \quad (7)$$

| Method | ImageNet | | Caltech101 | | OxfordPets | | StanfordCars | | Flowers102 | |
|---|---|---|---|---|---|---|---|---|---|---|
| | Base | Novel | Base | Novel | Base | Novel | Base | Novel | Base | Novel |
| CLIP | 72.43 | 68.14 | 96.84 | 94.00 | 91.17 | 97.26 | 63.37 | 74.89 | 72.08 | 77.80 |
| CoOp | 76.47 | 67.88 | 98.00 | 89.81 | 93.67 | 95.29 | 78.12 | 60.40 | 97.60 | 59.67 |
| Co-CoOp | 75.98 | 70.43 | 97.96 | 93.81 | 95.20 | 97.69 | 70.49 | 73.59 | 94.87 | 71.75 |
| ProGrad | 77.02 | 66.66 | 98.02 | 93.89 | 95.07 | 97.63 | 77.68 | 68.63 | 95.54 | 71.87 |
| KgCoOp | 75.83 | 69.96 | 97.72 | 94.39 | 94.65 | 97.76 | 71.76 | 75.04 | 95.00 | 74.73 |
| MaPLe | 76.66 | 70.54 | 97.74 | 94.36 | 95.43 | 97.76 | 72.94 | 74.00 | 95.92 | 72.46 |
| ΛPOLLO | **78.91** | **73.43** | **98.69** | **95.63** | **97.11** | **98.94** | **78.16** | **75.66** | **98.07** | **77.83** |
| $\Delta_{\text{ΛPOLLO}-\text{MaPLe}}$ | +2.25 | +2.89 | +0.95 | +1.27 | +1.68 | +1.18 | +5.22 | +1.66 | +2.15 | +5.37 |

| Method | Food101 | | FGVCAircraft | | SUN397 | | DTD | | UCF101 | |
|---|---|---|---|---|---|---|---|---|---|---|
| | Base | Novel | Base | Novel | Base | Novel | Base | Novel | Base | Novel |
| CLIP | 90.10 | 91.22 | 27.19 | 36.29 | 69.36 | 75.35 | 53.24 | 59.90 | 70.53 | 77.50 |
| CoOp | 88.33 | 82.26 | 40.44 | 22.30 | 80.60 | 65.89 | 79.44 | 41.18 | 84.69 | 56.05 |
| Co-CoOp | 90.70 | 91.29 | 33.41 | 23.71 | 79.74 | 76.86 | 77.01 | 56.00 | 82.33 | 73.45 |
| ProGrad | 90.37 | 89.59 | 40.54 | 27.57 | 81.26 | 74.17 | 77.35 | 52.35 | 84.33 | 74.94 |
| KgCoOp | 90.50 | 91.70 | 36.21 | 33.55 | 80.29 | 76.53 | 77.55 | 54.99 | 82.89 | 76.67 |
| MaPLe | 90.71 | 92.05 | 37.44 | 35.61 | 80.82 | 78.70 | 80.36 | 59.18 | 83.00 | 78.66 |
| ΛPOLLO | **91.35** | **92.58** | **40.62** | **39.87** | **82.97** | **80.62** | **83.95** | **65.21** | **87.88** | **80.52** |
| $\Delta_{\text{ΛPOLLO}-\text{MaPLe}}$ | +0.64 | +0.53 | +3.18 | +4.26 | +2.15 | +1.92 | +3.59 | +6.03 | +4.88 | +1.86 |

Table 1: **Comparison of** ΛPOLLO **with SOTA methods on a Base-to-Novel Class Generalization task.** ΛPOLLO shows strong generalization capabilities across all **10 datasets** and outperforms MaPLe (previous SOTA) in all of them. Best accuracy values are shown in bold and the differences with respect to MaPLe are given in blue.

## 4 Experimental Details

### 4.1 Downstream Tasks

**Base-to-Novel Class Generalization:** We follow the standard practice of base-to-novel generalization under zero-shot evaluation with few-shot fine-tuning. Datasets are partitioned into base and novel classes with the model being trained on the base classes and evaluated on both base and novel categories.

**Cross-Dataset Evaluation:** To analyze the zero-shot generalizability of ΛPOLLO , we perform a cross-dataset assessment. To facilitate this, the model trained solely on the ImageNet dataset was exposed to other 9 datasets. To be consistent with the prior art, we follow MaPLe and CoCoOp to train our model under a few-shot setting for a fair assessment.

**Domain Generalization:** Taking one step further towards a more robust evaluation, the performance of the model was analyzed on its out-of-distribution generalization capabilities. Like cross-dataset evaluation, the performance of the ImageNet trained model was observed under its four other variants: ImageNetV2, ImageNetS, ImageNetA, and ImageNetR as these are known to contain sufficient domain shifts.

### 4.2 Datasets

To extensively evaluate our model under different setting, we consider a total of **10 image classification datasets** covering various recognition tasks including object classification datasets ImageNet (Deng et al., 2009), Caltech 101 (Fei-Fei et al., 2004); fine-grained datasets Oxford-Pets (Parkhi et al., 2012), StanfordCars (Krause et al., 2013), Flowers 102 (Nilsback and Zisserman, 2008), Food101 (Bossard et al., 2014), FGV-CAAircraft (Maji et al., 2013); scene recognition dataset SUN397 (Xiao et al., 2010); action recognition dataset UCF101 (Soomro et al., 2012); and texture recognition dataset DTD (Cimpoi et al., 2014). For domain generalization, we rigorously evaluate on four ImageNet variants: ImageNetV2 (Recht et al., 2019), ImageNetSketch (Wang et al., 2019), ImageNet-A (Hendrycks et al., 2021b), and ImageNet-R (Hendrycks et al., 2021a).

### 4.3 Main Results

#### 4.3.1 Base-to-Novel Class Generalization

We subdivide the generalization evaluation into the following two categories.

**Generalization to Unseen Classes:** Table 1 presents the comparison of our method with CLIP, CoOp (Zhou et al., 2022b), CoCoOp (Zhou et al., 2022a), ProGrad (Zhu et al., 2022), KgCoOp (Yao

| Method | Target | | | | | | | | |
|---|---|---|---|---|---|---|---|---|---|
| | Caltech | Pets | Cars | Flowers | Food | Aircraft | SUN | DTD | UCF |
| CoOp | 93.70 | 89.14 | 64.51 | 68.71 | 85.30 | 18.47 | 64.15 | 41.92 | 66.55 |
| Co-CoOp | 94.43 | 90.14 | 65.32 | 71.88 | 86.06 | 22.94 | 67.36 | 45.73 | 68.21 |
| MaPLe | 93.53 | 90.49 | 65.57 | 72.23 | 86.20 | 24.74 | 67.01 | 46.49 | 68.69 |
| APOLLO | **95.12** | **91.56** | **66.21** | **73.15** | **86.82** | **24.96** | **67.98** | **47.63** | **70.38** |
| $\Delta_{\text{APOLLO}-\text{MaPLe}}$ | +1.59 | +1.07 | +0.64 | +0.92 | +0.62 | +0.22 | +0.97 | +1.14 | +1.69 |

Table 2: **Comparison of** APOLLO **with SOTA methods on a cross-dataset evaluation task** where the model is trained on ImageNet and evaluated on the target datasets in a zero-shot manner. APOLLO obtains the best accuracy among the existing methods suggesting better generalization capabilities. Best accuracies are presented in bold and improvements over MaPLe (previous SOTA) are shown in blue.

| Method | Target | | | |
|---|---|---|---|---|
| | ImNetV2 | ImNetS | ImNetA | ImNetR |
| CLIP | 60.83 | 46.15 | 47.77 | 73.96 |
| CoOp | 64.20 | 47.99 | 49.71 | 75.21 |
| Co-CoOp | 64.07 | 48.75 | 50.63 | 76.18 |
| ProGrad | 64.73 | 47.61 | 49.39 | 74.58 |
| KgCoOp | 64.10 | 48.97 | 50.69 | 76.70 |
| MaPLe | 64.07 | 49.15 | 50.90 | 76.98 |
| APOLLO | **64.89** | **50.17** | **51.74** | **78.33** |
| $\Delta_{\text{APOLLO}-\text{MaPLe}}$ | +0.82 | +1.02 | +0.84 | +1.35 |

Table 3: **Comparison of** APOLLO **with SOTA methods on a domain generalization task.** APOLLO demonstrates the best performance across all datasets. Best accuracies are present in bold and improvements over MaPLe (previous SOTA) are shown in blue.

et al., 2023), MaPLe on novel classes. Experimental results show the superiority of the proposed approach as it outperforms all existing methods by a significant margin on zero-shot generalization. It achieves a relative gain of up to 6.03% over the most recent baseline. Evaluating on all 10 datasets, we find an average improvement of 2.69% over MaPLe on novel categories. As is evident from the tables none of the existing methods beat pre-trained CLIP (except MaPLe) underlining the challenges of achieving satisfactory zero-shot performance while learning a new task in a few-shot setting.

Our method also enjoys an average improvement of 2.80% over CLIP on novel classes across all the datasets. This can be attributed to our unified multi-modal adapter and prompting technique that enables the model to leverage the mutual association between visual and language modalities.

**Generalization and Performance on Base Classes:** CoCoOp extends CoOp by introducing image-conditioned prompts thus securing considerable generalizability and obtaining a significant performance boost over the latter. The downside is unsatisfactory performance on base classes as it is only effective on 2 / 10 datasets with a large drop in average performance. MapLe on the other hand betters CoCoOp on most of the datasets (Table 1). Note that CoOp lacks generalizability over novel classes due to its massive overfitting with the base classes. In contrast, APOLLO alleviates this limitation and sees a large gain over its predecessors proving its effectiveness in learning new classes while maintaining a stable performance on base classes. Note that, our method outperforms the current benchmark over all 10 datasets to obtain few-shot performance improvements up to 5.22% on base categories and 2.66% overall. This suggests that the gain under a zero-shot setting does not affect its few-shot performance or the other way around.

### 4.3.2 Cross-Dataset Evaluation

Table 2 presents the results on a cross-dataset evaluation where the model is trained solely on ImageNet and evaluated on other 9 datasets under the zero-shot setting. We find APOLLO surpasses all other methods and observe the best zero-shot improvement of 1.69% over MaPLe (SOTA) on the UCF dataset.

### 4.3.3 Domain Generalization

Table 3 illustrates domain generalization results of APOLLO . We use the ImageNet dataset as the base and test our model on ImageNetV2, ImageNetS, ImageNetA, and ImageNetR where all of them are from vastly different distributions. We notice strong generalizability across different domains as APOLLO outperforms all prior baselines by achieving the best zero-shot improvement of 1.35% over MaPLe on the ImageNetR dataset.

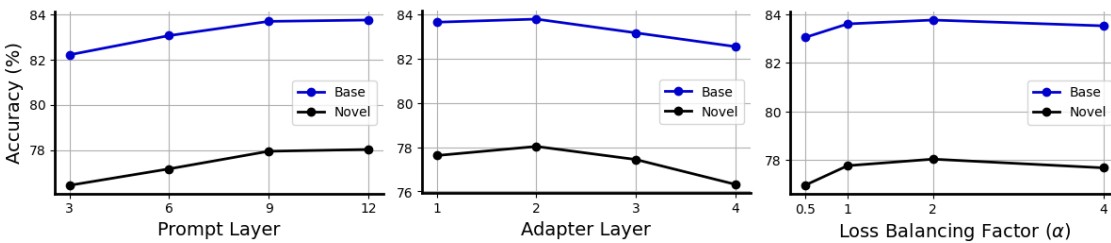

Figure 3: Impact of Prompt Layer, Adapter Layer, and Loss Balancing Factor ($\alpha$) on the performance of APOLLO .

| Intra-modal | Adap. | Cross-attn. | Avg.(Base) | Avg.(Novel) |
|:---:|:---:|:---:|:---:|:---:|
| ✗ | ✗ | ✗ | 81.49 | 75.62 |
| ✓ | ✗ | ✗ | 82.36 | 76.45 |
| ✓ | ✓ | ✗ | 83.05 | 77.22 |
| ✗ | ✓ | ✗ | 82.30 | 76.21 |
| ✗ | ✓ | ✓ | 82.93 | 77.07 |
| ✓ | ✓ | ✓ | **83.77** | **78.03** |

Table 4: **Impact of Intra-modal contrastive consistency, Adapter tuning, and Cross-attention strategies** on the performance of APOLLO . Our method gives the best accuracy values (averaged on 10 datasets for base-to-novel generalization task) when all three components are considered.

## 4.4 Ablations

**Impact of Intra-modal Contrastive Consistency, Adapter, and Cross-attention:** We assess the importance of the individual components in APOLLO and report the average accuracy scores (on 10 datasets) for the Base-to-Novel Class Generalization task (see Table 4). We show 3 main components in APOLLO , i.e., Intra-modal Contrastive Consistency, Adapter Layers, and the utility of the Cross-attention strategy. Note that Prompt Learning is present in all these experiments. Furthermore, input augmentation is present in cases where intra-modal consistency is taken into account. We observe the lowest performance with only prompt learning, shown in ($1^{st}$ row) in Table 4. Upon addition of intra-modal contrastive consistency, we notice a significant boost (+0.87% in base and +0.83 % in novel) in performance. Therefore, enforcing the mutual synergy between the same modalities results in a much better generalization across several datasets. Intra-modal consistency when paired with an adapter gives a further improvement (+0.69% in base and +0.77 % in novel) in performance suggesting that they mutually aid each other in the generalization task. Finally, when cross attention is employed in the adapter layers we obtain +0.63 % and +0.86 % relative improvements in the base and novel classes respectively.

**Impact of Prompt Layers:** We evaluate the average performance of APOLLO on 10 datasets for the base-to-novel generalization with respect to the prompt depth, i.e., the number of prompt layers (see Figure 3). Unlike MaPLe (Khattak et al., 2023), while conducting ablation, we consider the same depth (num layers) in both vision and language encoders. The general trend suggests that with increased prompt depth, the overall performance improves. Further, unlike MaPLe, we obtain a slight improvement in accuracy values for the generalization task when prompt depth is increased from 9 to 12. The reason for such an observation can be attributed to the presence of adapter layers and intra-modal contrastive consistency losses which negatively impact performance related to increasing prompt depth beyond 9. Therefore, we maintain the value of prompt depth to 12 in all our experiments.

**Impact of Adapter Layers:** To evaluate the importance of adapters we employ a dual setup strategy. **First**, we study the utility of adapters in individual modalities (see Appendix Table 8). We notice that adding a text adapter is more beneficial than adding only an image adapter. This is consistent with the findings outlined in Gao et al. (2021c). However, unlike them, we notice an increase in accuracy values when adapters are used with both modalities. A key difference in this respect (as compared to Gao et al. (2021c)) is the presence of cross-attention which plays an important role in alignment. This **demonstrates the importance of** yet another component of APOLLO , i.e., the **intra-modal contrastive consistency**. This underlines that dual-modality adapters are not beneficial in the case of naive few-shot tuning, whereas our approach (having both cross-attention and intra-modal consistency) leads to better generalization with more tunable parameters on both modalities. **Second**, we study the role of the number of adapter layers using 4 different configurations as shown in

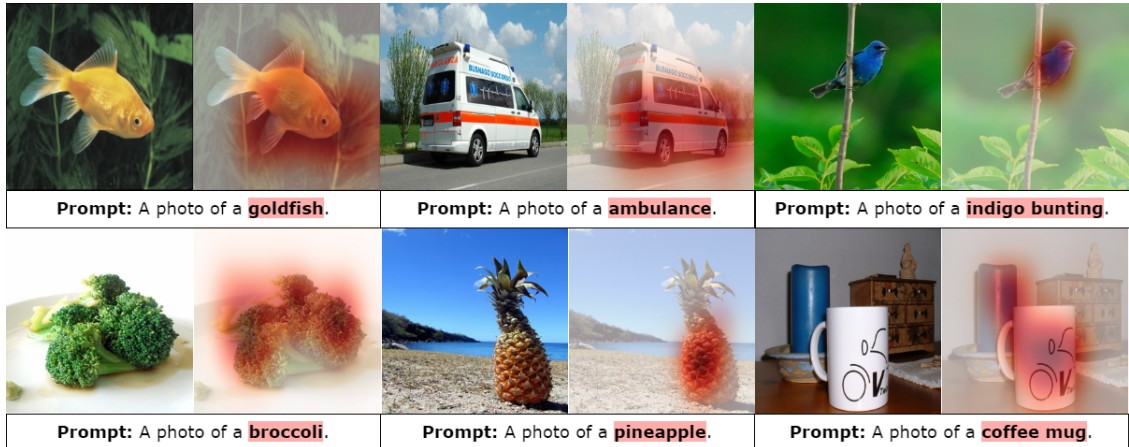

Figure 4: **Cross-attention visualizations as heatmaps superimposed on the respective original images** showing how objects (in red) in text prompts attend to relevant regions in the images. These maps depict the cross-modal alignment which improves generalization in downstream tasks.

Figure 3. Note that increasing adapter depth from 1 to 2 leads to an improvement in performance indicating its role in understanding more complex relationships between the two modalities via cross-modal alignment. However, the performance drops beyond the depth of 2 suggesting that overparameterization may lead to an overfitting issue under a few-shot setting.

**Impact of Input Augmentations:** Augmentation plays a pivotal role in enforcing consistency across different branches (Chen et al., 2020; Zbontar et al., 2021). For text augmentation, we use two strategies – Easy Data Augmentation (Wei and Zou, 2019) and LLM-generated augmentation. We show that using a more descriptive text of the object employing LLM is more beneficial for learning intra-modal consistency than using the EDA-based approach (see Table 9). For image augmentation, we consider two schemes – standard image augmentations such as cropping, flipping, solarization, and color inversion, and text-conditioned diffusion-based augmentation. We observe that the latter performs better in our case (Table 9).

**Impact of Loss Balancing Factor:** To study the impact of the Loss Balancing (Weighting) Factor ($\alpha$), we report the average accuracy values (on 10 datasets) of Base and Novel classes in the Base-to-Novel Generalization task. As shown in Figure 3, an increase in the value of $\alpha$ leads to an increased weightage of intra-modal loss compared to the inter-modal similarity loss – this leads to an improvement in performance. We achieve optimal results for $\alpha = 2$ which we use in all the experiments. However, an over-emphasis on the

intra-modal loss leads to poor cross-modal alignment affecting the overall performance.

**Different LLMs as Text Augmentors** Table 5 reports the performances of ApoLLo under different LLM-based text augmenters.

| LLM Text Augmentor | Avg.(Base) | Avg.(Novel) |
|---|---|---|
| VICUNA | 83.16 | 77.29 |
| **GPT** | **83.77** | **78.03** |

Table 5: **Performance comparison between different LLM-based text augmenters.**

**Adapters on Different Image and Text Branches** Our method contains two image and text branches (where Image Branch 1, Image Branch 2, Text Branch 1, and Text Branch 2 are in order from left to right in Figure 2 and represent the augmented image, original image, original text, and augmented text respectively). Here, we add the following table which demonstrates the effects of adapter layers on different image and text branch combinations. Please note that all these combinations involve both self- and cross-attentions. We obtain similar accuracy values in two cases: (a) Image Branch 2 - Text Branch 1 combination, and (b) when all four branches are taken into account. However, we select case (a) with Image Branch 2 and Text Branch 1 combination because the number of trainable parameters in this scenario is half that in the latter. Moreover, Image Branch 2 - Text Branch 1 combination of adapters gives the highest (mean) accuracy values on base and novel classes as shown in Table 6.

| Img. Br. 1 | Img. Br. 2 | Txt. Br. 1 | Txt. Br. 2 | Avg.(Base) | Avg.(Novel) |
|:---:|:---:|:---:|:---:|:---:|:---:|
| ✓ | ✗ | ✓ | ✗ | 83.28 | 77.32 |
| ✗ | ✓ | ✗ | ✓ | 83.19 | 77.24 |
| ✓ | ✗ | ✗ | ✓ | 83.16 | 77.23 |
| ✓ | ✓ | ✓ | ✓ | **83.77** | 78.01 |
| ✗ | ✓ | ✓ | ✗ | **83.77** | **78.03** |

Table 6: **Impact of adding adapters on different image and text branches.**

**Incorporating Attentions in Adapter Layers** We add the following ablations in Table 7 w.r.t the Adapter layers: (a) Adapter layers having no attention (b) Adapter layers with only self-attention (c) Adapter layers with self + cross attention. Experimental results in Table 7) demonstrate optimal performance is achieved in (c) when cross attention is employed between the adapters leading to better multi-modal alignment.

| No-attn. | Self-attn. | Cross-attn. | Avg.(Base) | Avg.(Novel) |
|:---:|:---:|:---:|:---:|:---:|
| ✓ | ✗ | ✗ | 82.54 | 76.98 |
| ✗ | ✓ | ✗ | 83.05 | 77.22 |
| ✗ | ✓ | ✓ | **83.77** | **78.03** |

Table 7: **Impact of incorporating attention in adapter layers.**

### 4.5 Qualitative Results

Through Figure 4 and Appendix Figure 6 we visualize cross-attention maps guided by the respective text prompts. The cross-attention scores between the image and the associated text token referring to the object (highlighted in red in each prompt) are extracted and bilinearly interpolated to match the image dimension and superimposed on the original image. These maps depict the alignment between images and corresponding texts for a variety of classes, including fish, food, and vehicles. Such cross-modal awareness leads to better downstream tasks.

**Qualitative Analysis on Multi-object Setting** Examples of the cross-attention visualization on images with multiple objects are shown in Figure 5. Here, each of the images contains more than one object category. We generate textual prompts using the same template 'A photo of a <CLASS>' for individual objects as shown in the figure. The attention maps are activated in the corresponding regions showing fine-grained alignment of APoLLo under challenging scenarios with objects from multiple categories. Please note that here we did a zero-shot transfer on these images with the model trained on ImageNet using our method.

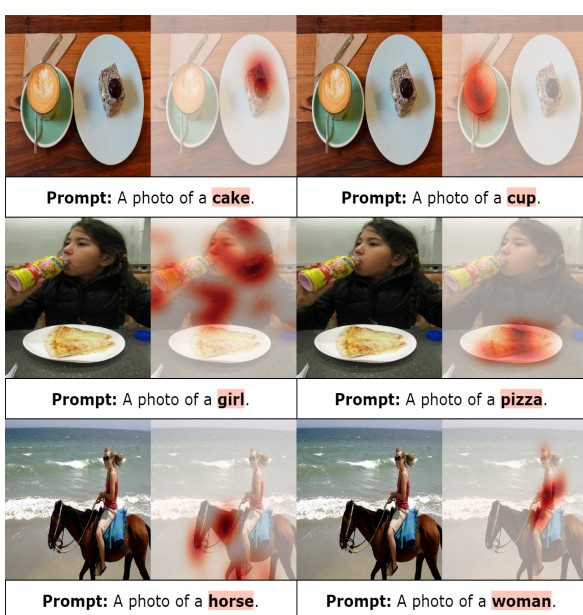

Figure 5: **Cross-attention visualizations as heatmaps superimposed on the respective original images in a multi-object setting**.

## 5 Conclusions and Future Work

We present APOLLO, a unified multi-modal adapter and prompt tuning method to facilitate few-shot learning without compromising zero-shot generalizability. To this end, we leverage contrastive-consistency loss and ensure cross-modal synergy which results in an improved performance on three carefully chosen downstream tasks: base-to-novel generalization, cross-dataset evaluation, and domain generalization on 10 benchmark datasets as observed from the experimental results. Moreover, the ablation analysis outlines the salient contributions of each of the sub-modules. Future work can include assessing the performance of such frameworks for fine-grained tasks which remains an open problem. Moreover, the adapter layers may lead to over-parameterization which can be further optimized in subsequent works.

**Acknowledgement:** The research has been supported by ARO grants W911NF2310352 and W911NF2110026.

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

# A  Appendix

## A.1  Details for Radar Chart Figure 1

In this section, we explain the details of the radar chart shown in Figure 1. Each axis denotes an accuracy value, either on base or novel class (specified accordingly) for the Base-to-Novel Generalization task. In total, there are 10 datasets for this task and therefore we plot 10 vertices where each denotes a ratio relative to our performance. This is calculated on the basis of normalizing the performance of either $\mathcal{A}$POLLO or previous SOTA (i.e., MaPLe) by that of $\mathcal{A}$POLLO , and is therefore kept in the range of (0, 1]. Here we set the radar chart's origin to be 90% and the outermost frame to be 100%. This essentially separates the adjacent frames for better readability. The accuracy values annotated against the vertices are the absolute values (without normalization) obtained using both methods.

## A.2  Intra-modal Contrastive Consistency Loss

Our intra-modal contrastive consistency loss differs from a conventional contrastive loss in the aspect that it contains consistency terms that prevent **class-collision** problem (Wei et al., 2020). Therefore, it has two terms - a contrastive loss and a consistency loss.

### A.2.1  Contrastive Loss

A typical contrastive loss function in the form of an InfoNCE loss is given as:

$$\mathcal{L}_{NCE} = -\log \frac{\exp(\mathbf{q} \cdot \mathbf{p}/\tau)}{\exp(\mathbf{q} \cdot \mathbf{p}/\tau) + \sum_{\mathbf{n}_k} \exp(\mathbf{q} \cdot \mathbf{n}_k/\tau)}$$

$$(8)$$

where $\mathbf{p}$ represents the positive key, $\mathbf{q}$ represents the query, and $\mathbf{n}_k$ represents the negative key in a minibatch.

### A.2.2  Consistency Loss

Taking inspiration from semi-supervised learning, consistency loss is proposed by Wei et al. (2020) to strengthen the consistency between the similarities of the query data and the positive data.

The similarity between the query $\mathbf{q}$ and the negative keys $\mathbf{n}_k$ can be represented in the form of a probability $\mathcal{Q}(i)$ which is denoted as:

$$\mathcal{Q}(i) = \frac{\exp(\mathbf{q} \cdot \mathbf{n}_i/\tau_{con})}{\sum_{\mathbf{n}_k} \exp(\mathbf{q} \cdot \mathbf{n}_k/\tau_{con})}$$

$$(9)$$

The similarity between the positive $\mathbf{p}$ and the negative keys $\mathbf{n}_i (i \in \{1, 2, \ldots, K\})$ in the form of a probability $\mathcal{P}(i)$ is written as:

$$\mathcal{P}(i) = \frac{\exp(\mathbf{p} \cdot \mathbf{n}_i/\tau_{con})}{\sum_{\mathbf{n}_k} \exp(\mathbf{p} \cdot \mathbf{n}_k/\tau_{con})}$$

$$(10)$$

Consistency between the probability distributions $\mathcal{P}$ and $\mathcal{Q}$ is imposed in the form of a Symmetric KL Divergence Loss.

$$\mathcal{L}_{consistency} = \frac{1}{2}(KL(\mathcal{P}, \mathcal{Q}) + KL(\mathcal{Q}, \mathcal{P}))$$

$$(11)$$

The predicted similarity distribution of the positive key to each crop of the other data, $\mathcal{P}$, acts as a soft pseudo label to that of the query, $\mathcal{Q}$. The total loss is a weighted combination of contrastive and consistency losses given as:

$$\mathcal{L}_{CO_2} = \mathcal{L}_{NCE} + \beta \mathcal{L}_{consistency}$$

$$(12)$$

where $\beta$ is the balancing coefficient. We follow the recommended values of $\tau_{con}$ and $\beta$ as mentioned in Wei et al. (2020) since they have shown to perform the best in our experiments.

## A.3  Inter-modal Similarity Maximization

During training we consider a mini-batch ($\mathcal{N}$) containing image-captions pairs, $\{I_j, T_j\}_{j=1}^{\mathcal{N}}$, where $I_j$ and $T_j$ represent $j^{th}$ image and text pair, respectively. After passing these image-text pairs through respective encoders (and adapter layers) we obtain the normalized image embedding as $z_j^I \in \mathbb{R}^d$ and text embedding as $z_j^T \in \mathbb{R}^d$. These representations are also **aware** of each other (modality-wise). They are subsequently used to compute this inter-modality Image-Text Contrastive (ITC) loss given as $\mathcal{L}_{sim}^{IT}$.

$$\mathcal{L}_{sim}^{IT} = -\frac{1}{2\mathcal{N}} \sum_{j=1}^{\mathcal{N}} \log \underbrace{\left[ \frac{\exp\left(\langle z_j^I, z_j^T \rangle/\tau\right)}{\sum_{l=1}^{\mathcal{N}} \exp\left(\langle z_j^I, z_l^T \rangle/\tau\right)} \right]}_{\text{Contrasting images with the texts}}$$

$$-\frac{1}{2\mathcal{N}} \sum_{l=1}^{\mathcal{N}} \log \underbrace{\left[ \frac{\exp\left(\langle z_l^I, z_l^T \rangle/\tau\right)}{\sum_{j=1}^{\mathcal{N}} \exp\left(\langle z_j^I, z_l^T \rangle/\tau\right)} \right]}_{\text{Contrasting texts with the images}}$$

$$(13)$$

where $\langle \cdot, \cdot \rangle$ denotes inner product, and $\tau$ is the temperature parameter.

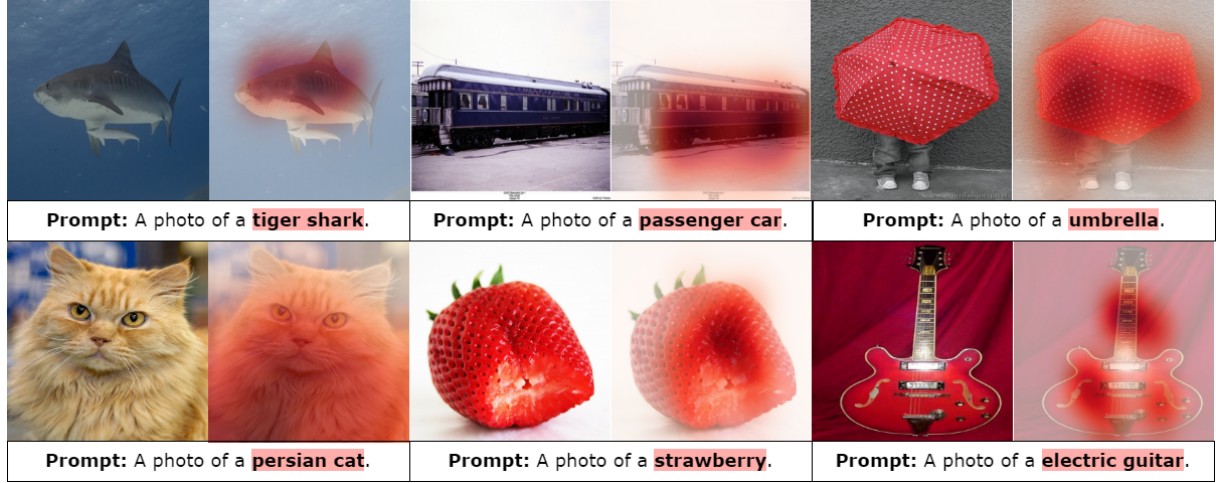

Figure 6: **Cross-attention visualizations as heatmaps superimposed on the respective original images** showing how objects (in red) in text prompts attend to relevant regions in the images (Extension of Figure 4).

## A.4 Generative Data Augmentation

Generative data augmentation is an emerging field where generative models are used to generate augmented (perturbed) samples of the data (Hu et al., 2023; Trabucco et al., 2023; Azizi et al., 2023; Shipard et al., 2023; Yin et al., 2023; Whitehouse et al., 2023). It is prominent in the field of Natural Language Processing where Large Language Models (LLMs) are used as data augmenters (Whitehouse et al., 2023; Yin et al., 2023). Given a prompt, LLMs provide several descriptive versions of it which can potentially be used to increase the size of the dataset, especially in a low-dataset regime. Further, text-to-image generation and diffusion models (Yin et al., 2023; Trabucco et al., 2023; Azizi et al., 2023; Shipard et al., 2023) are also used for effective data augmentation strategies. However, these types of augmentation schemes have not been explored together previously in a multi-modal (vision-language) context. To the best of our knowledge, ours is the first to leverage the capabilities of LLMs and text-conditioned image generation (diffusion) models together in a unified framework for effective augmentation in a multi-modal learning context.

## A.5 Implementation Details

Following MaPLe (Khattak et al., 2023), we utilize the pre-trained ViT-B/16 CLIP model as our vision-language backbone. In all experiments, we fine-tune the model employing a few-shot training strategy with 16 samples randomly sampled from each class for all known classes. We train our model on a single GPU for 10 epochs using an SGD optimizer with a base learning rate of 0.004. We follow MaPLe for setting the values of prompt depths and lengths in our experiments.

## A.6 Ablation Tables

### A.6.1 Impact of Adapter Layers

The impact of adapter layers is shown in Table 8.

| Image | Text | Average | |
|:---:|:---:|:---:|:---:|
| | | Base | Novel |
| ✓ | ✗ | 83.16 | 77.31 |
| ✗ | ✓ | 83.41 | 77.48 |
| ✓ | ✓ | 83.77 | 78.03 |

Table 8: **Impact of using different adapters** on the performance of APOLLO . Our method gives the best accuracy values when both image and language adapters are used.

### A.6.2 Impact of Input Augmentations

Impact of input augmentation is shown in Table 9.

| Modality | Augmentation | Average | |
|:---:|:---:|:---:|:---:|
| | | Base | Novel |
| Image | Standard | 83.28 | 77.45 |
| | Diffusion | 83.77 | 78.03 |
| Text | EDA | 82.81 | 77.04 |
| | LLM | 83.77 | 78.03 |

Table 9: **Impact of different input augmentation strategies** on the performance of APOLLO . Our method gives the best accuracy values when LLM is used for text augmentation and the text-condition image generation model is used for image augmentation.