# OpenReview forum: "APoLLo : Unified Adapter and Prompt Learning for Vision Language Models"
_EMNLP/2023/Conference — EMNLP 2023 Main_

### Official Review · Reviewer_SvZA · 2023-08-04

**Soundness:** 3

**Excitement:**

3: Ambivalent: It has merits (e.g., it reports state-of-the-art results, the idea is nice), but there are key weaknesses (e.g., it describes incremental work), and it can significantly benefit from another round of revision. However, I won't object to accepting it if my co-reviewers champion it.

**Paper Topic And Main Contributions:**

This paper presents an interesting framework called APoLLo. The authors propose a method that includes both prompt tuning and adapter tuning in the scenario of multimodalities.
Following the previous works, the authors use the classic ITC and a consistency-based contrastive loss for training.
THe authors demonstrate good results on three selected downstream tasks.

**Questions For The Authors:**

- Could you please provide some statistics for the training cost? I wonder if the augmented images are generated offline to accelerate training. If both the LLM and diffusion models augment data on the fly, it seems very expensive in terms of a single GPU.
- The augmentation greatly depends on the diffusion and LLM models. It may be helpful to report performance impact with different size of diffusion/LLM models. This may point out the hard requirements for those models.

**Reasons To Accept:**

- The designed network architecture is interesting.
- The proposed method achieves strong improvements on three categories of downstream tasks.
- The authors provide a large number of empirical results and interesting discussions.

**Reasons To Reject:**

- The submission is more like an empirical/engineering study. While the developed model and training recipe are greatly appreciated, the performance improvements are straightforward and expected, mainly attributed to the training data augmentation. In other words, there is limited novelty in the presented method. Both the loss functions are explored in prior works. LLM-based and text-conditioned augmentation are also well-explored. The only new thing in the paper is to combine the two. But this improves little performance as shown in Table A.6.1 and A.6.2.
- As the diffusion model and LLM need to be well pre-trained with a variety of different datasets, I am not sure if the reported experiments are of fair comparison.
- Visualization in Fig 4 is not convincing. For each demonstrated example, it only has a single salient object in the image and the text is of a template "A photo of a [OBJ]". Under the well-restricted setting, it is expected to see attention around the salient object. It is not conclusive to validate the effectiveness of cross-modal alignment.

**Reproducibility:**

2: Would be hard pressed to reproduce the results. The contribution depends on data that are simply not available outside the author's institution or consortium; not enough details are provided.

**Reviewer Confidence:**

4: Quite sure. I tried to check the important points carefully. It's unlikely, though conceivable, that I missed something that should affect my ratings.

---

> ### Author Rebuttal · Authors · 2023-08-25
>
> We thank the reviewer for the thoughtful and constructive comments. Please find our responses below. The tables here represent mean accuracies reported over all the 10 datasets.
>
> $\textbf{1. Novelties, contributions and performance gains}$
>
> We would like to mention that one of our primary contributions is the unification of adapter and prompt learning within the same framework for VLP models (Section 3.2). To the best of our knowledge, this has not been explored by the community before. Our method facilitates learning new tasks in a few-shot setting without compromising on their zero-shot generalizability.
>
> Please refer to Table 1 of the main paper which shows the superior performance of ApoLLo over prior SOTA MaPLe (Khattak et. al. 2023). Note that the highest performance gain of $\textbf{6.03%}$ is obtained in Novel classes of DTD dataset. To show the overall efficacy of our method we report in Table A.6.1 and A.6.2 the average (over 10 diverse datasets) accuracy improvements. It is to be noted that, as outlined in CLIP (Radford et al. 2021), there is less room for improvements under zero-shot setting in some of the datasets. Similar trends in the results and corresponding improvements have also been found in studies such as MaPLe, KgCoOp (Yao et. al 2023). Therefore, reporting the mean of all the datasets (as shown in Table A.6.1 and A.6.2), does not fully capture those cases where we obtain significant improvements on the individual datasets (refer Table 1). However, for ablation studies we report the mean accuracy values following previous works for fairness (MaPLe).
>
> In addition to this as outlined in the paper (under main results in page 2), we make the following novel contributions:
>
> (a) We propose a novel multi-modal augmentation strategy by leveraging LLMs to generate descriptive texts as augmented samples in the text branch, and text-conditioned diffusion models to generate image augmentations for the image branch. Although LLM based augmentations have been explored before, to the best of our knowledge, multi-modal augmentation is relatively unexplored.
>
> (b) Our novel application of multi-modal cross attention adapter layers bridges the gap between the two modalities by generating text-guided visual  features and vice-versa. This promotes the synergy  between the two modalities.
>
> (c) Finally, extensive evaluation on 10 challenging datasets demonstrates the effectiveness of APoLLo as it outperforms existing methods by a significant margin and set a new SOTA (Figure 1) for a range of downstream tasks including base-to-novel generalization, cross-dataset recognition, and domain generalization.
>
> $\textbf{2. Applicability of diffusion and LLMs as data augmentors}$
>
> Generative data augmentation is an emerging field where generative models are used to generate augmented (perturbed) samples of the data (Hu et al. 2023; Trabucco et al., 2023; Azizi et al., 2023; Shipard et al. 2023; Yin et al., 2023; Whitehouse et al. 2023,). It is prominent in the field of Natural Language Processing where Large Language Models (LLMs) are used as data augmenters (White house et al., 2023; Yin et al. 2023,). Given a prompt, LLMs provide several descriptive versions of it which can potentially be used to increase the size of the dataset, especially in a low-dataset regime. Further, text-to-image generation and diffusion models (Yin et al.,2023; Trabucco et al.,2023; Azizi et al.,2023; Shipard et al.2023,) are also used for effective data augmentation strategies.
>
> Hence it is a common practice to use LLMs as augmentors by leveraging the capabilities of these models which are trained on varied datasets. Therefore we believe using them effectively as a tool for data augmentation in few-shot settings can boost the performance (as mentioned in Table A.6.2).
>
> $\textbf{3. Cross-attention visualization on images with multiple objects}$
>
> Thank you for bringing this to our notice. As suggested by the reviewers, we provide some examples of the cross-attention visualization on images (from COCO validation set) with multiple objects embedded in the anonymous link (as approved by the program chairs): https://ibb.co/dczvjcN
>
> Here, each of the images contains more than one object category. We generate textual prompts using the same template “A photo of a <CLASS>” for individual objects as shown in the figure. The attention maps are activated in the corresponding regions showing fine grained alignment of APoLLo under challenging scenarios with objects from multiple categories. Please note that here we did a zero-shot transfer on these images with the model trained on ImageNet using our method.
>
> $\textbf{4. Implementation details}$
>
> In our current set up, we generate the images using a text-to-image stable diffusion model a priori. On the other hand, the text augmentor was leveraged to generate on-the-fly text augmentations to ensure diversification in the generated samples.
>
> Experimental results show that this strategy yields superior performance in terms of computational resources as compared to when both the image and text augmentations are done offline. We also observe better model performances in the former case. We attribute this to the fact that when we use the LLM on-the-fly to obtain the textual augmentations they are able to generate diverse captions instead of a fixed caption (when done offline). To ensure the generated images are regulated within a certain variability (using FID scores), we make the image augmentation offline. Furthermore, for all our experiments, we have used a single A100 GPU and adopted memory efficient techniques such as gradient checkpointing as commonly used in the deep learning community.
>
> $\textbf{5. Impact of different text augmentors}$
>
> As also suggested by Reviewer $\textcolor{Green}{\text{bnbq}}$, we add the ablation studies with different diffusion based image augmentors and LLM based text augmentors. Following table reports the performances of ApoLLo under different LLM-based text augmentors. We will add these results in the Supplementary section of our paper.
>
> $\textbf{Table 3. Performance comparison between different LLM based text augmentors}$
>
> | LLM Text Augmentor | Avg. (Base) | Avg. (Novel) |
> | :----: | :----: | :-----: |
> | VICUNA  | 83.16 | 77.29 |
> | **GPT** | **83.77** | **78.03** |
>
>
> Also, as mentioned in the paper, we will make the codebase public upon acceptance to facilitate further research in this direction.
>
> $\textbf{References}$
>
> MaPLe: Multi-modal prompt learning (Khattak et. al 2023)
>
> Learning Transferable Visual Models From Natural Language Supervision (Radford et. al. 2021)
>
> Visual-language prompt tuning with knowledge-guided context optimization (Yao et. al, 2023)
>
> Gda: Generative data augmentation techniques for relation extraction tasks (Hu et. al. 2023)
>
> Effective data augmentation with diffusion models (Trabucco et. al. 2023)
>
> Synthetic data from diffusion models improves imagenet classification. (Azizi et. al. 2023)
>
> LLM-powered Data Augmentation for Enhanced Crosslingual Performance (Whitehouse et al. 2023)
>
> TTIDA: Controllable Generative Data Augmentation via Text-to-Text and Text-to-Image Models (Yin et al. 2023)
>
> Diversity is definitely needed: Improving model-agnostic zero shot classification via stable diffusion (Shipard et. al. 2023)

---

### Official Review · Reviewer_4n34 · 2023-08-05

**Soundness:** 4

**Excitement:**

4: Strong: This paper deepens the understanding of some phenomenon or lowers the barriers to an existing research direction.

**Paper Topic And Main Contributions:**

This paper proposes APoLLo, a fine-tuning method for visual-language pretrained (VLP) models for multi-modal downstream tasks. The proposed method is a novel integrated adapter and prompt learning approach for VLP models, addressing the issue of generalization in few-shot scenarios. The main contributions of this paper are as follows:

- The proposed method is the first to unify adapter tuning and prompt tuning for VLP models, facilitating the learning of new tasks in a few-shot setting without compromising zero-shot generalization.
- By leveraging LLMs to generate explanatory text for augmented samples and using text-conditioned diffusion models for image augmentation, a new multi-modal augmentation strategy is proposed.
- A novel application of multi-modal cross-attention adapter layers is introduced, bridging the gap between the two modalities and fostering a synergistic effect between them.
- Through evaluation on ten datasets, the effectiveness of APoLLo is demonstrated, setting new state-of-the-art benchmarks on various downstream tasks, including generalization from base to novel, cross-dataset recognition, and domain generalization.

**Questions For The Authors:**

- Which LLM and diffusion models were used in the paper?
- Is it one generated image per sample?
- Have evaluations been conducted on the generated images and text?

**Reasons To Accept:**

- The proposed method is the first to unify adapter tuning and prompt tuning, facilitating the learning of new tasks in a few-shot setting without compromising zero-shot generalization. By clarifying the differences from recent related research, the position of this study is clearly defined. Through comparison with the latest models on ten datasets, the utility of the method is demonstrated by its superior results. This contribution greatly contributes to the advancement of the related fields, yielding highly useful results.
- With the introduction of multi-modal augmentation via LLMs and text-conditioned diffusion models, the adoption of multi-modal cross-attention adapter layers, and the introduction of Intra-modal Contrastive Consistency, the impact of these components is clearly revealed through appropriate ablation studies. Furthermore, by visualizing cross-attention, the understanding of the proposed method is promoted, reflecting the exceptional quality of this paper.

**Reasons To Reject:**

- One issue that seems to remain is the lack of clarity regarding the impact of the quality and quantity of augmentations generated by the LLM and the text-conditioned diffusion model on the proposed method. The paper has demonstrated through experiments that input augmentation was effective, but there is a lack of detailed descriptions about the LLM and diffusion models used in the paper. It seems necessary to address the matter of their quality and quantity.

**Reproducibility:**

3: Could reproduce the results with some difficulty. The settings of parameters are underspecified or subjectively determined; the training/evaluation data are not widely available.

**Reviewer Confidence:**

4: Quite sure. I tried to check the important points carefully. It's unlikely, though conceivable, that I missed something that should affect my ratings.

**Typos Grammar Style And Presentation Improvements:**

- p.5 l366: A period is missing.

---

> ### Author Rebuttal · Authors · 2023-08-25
>
> We thank the reviewer for the thoughtful and constructive comments. Please find our responses below.
>
> $\textbf{1. Quality of augmentations}$
>
> Quality of generated data in an augmentation pipeline has been previously examined in (Whitehouse et al., 2023, Yin et. al. 2023, Peng et al. 2023,). Whitehouse et al. (2023) demonstrated the superiority of GPT variants over models such as VICUNA (Wei-Lin et al. 2023) as LLM-based text-augmentor using a Generation Success Rate metric. On the other hand, the quality of generated augmented samples have also been explored by Azizi et al. (2023) using FID scores. Taking inspiration from these works we considered pre-trained Stable Diffusion (Rombach et al. 2022) as the image augmentor and GPT as the text augmentor generating diverse yet meaningful image-text pairs. Furthermore, it is worth noting that the standard (non-generative) techniques of image data augmentation restrict themselves to simple transformations like rotations, flipping etc. However, these augmented images are not semantically diverse enough. In contrast, our method edits images to change their appearances using an off-the-shelf diffusion model, and generalizes to novel visual concepts from only a handful of samples.
>
> To the best of our knowledge, this is the first work to leverage the potential of these pretrained generative frameworks for data augmentation in a few-shot unified multimodal prompt and adapter learning setting. Please note quality assessment of the generated augmentations is beyond the scope of this work (can be a possible future work) and we conform to these prior works (mentioned before) which concludes that the augmentation strategies are indeed trust-worthy.
>
> $\textbf{2.  LLM and diffusion models}$
>
> As addressed above, the image augmentation branch consists of a pre-trained text-to-image stable diffusion model (Rombach et al. 2022) which is applied on input images to generate diverse yet meaningful image samples. The text branch comprises a pre-trained LLM, GPT (Brown et al. 2020), to obtain a descriptive sentence from an input prompt text of a generic format (a photo of a ‘class’).
>
> $\textbf{3. Number of generated images per sample}$
>
> Yes, for each input image-text pair we obtain a corresponding augmented image-text counterpart.
>
> $\textbf{4. Evaluations protocol}$
>
> Please note that the evaluations are not conducted on generated images and text due to the lack of a baseline performance of any existing model. For a fair comparison against all prior works and to be consistent with the datasets used in our work (see Section 4.2), we strictly use the original text and images for all our evaluations.
>
> $\textbf{4. Typos}$
>
> Thank you, we will update this in the camera ready version.
>
> Also, as mentioned in the paper, we will make all our code public upon acceptance to facilitate reproducibility.
>
> $\textbf{References}$
>
> LLM-powered Data Augmentation for Enhanced Crosslingual Performance (Whitehouse et al. 2023)
>
> TTIDA: Controllable Generative Data Augmentation via Text-to-Text and Text-to-Image Models (Yin et al. 2023)
>
> Check Your Facts and Try Again: Improving Large Language Models with External Knowledge and Automated Feedback (Peng et. al. 2023)
>
> Vicuna: An Open-Source Chatbot Impressing GPT-4 with 90\%* ChatGPT Quality (Wei-Lin et al. 2023)
>
> Language models are few-shot learners (Brown et al. 2020)
>
> High resolution image synthesis with latent diffusion models. (Rombach et al. 2022)

---

### Official Review · Reviewer_bnbq · 2023-08-11

**Soundness:** 4

**Excitement:**

4: Strong: This paper deepens the understanding of some phenomenon or lowers the barriers to an existing research direction.

**Paper Topic And Main Contributions:**

The authors propose APoLLo, which combines adapter and prompt learning to improve the generalization of these models when they are applied to new tasks with few examples. APoLLo introduces several novel aspects, such as a multi-modal augmentation strategy that generates descriptive texts and image augmentations using large language models (LLMs) and text-conditioned diffusion models. It also employs a contrastive-consistency loss that enforces consistency between the respective encoder branches and prevents overfitting in downstream tasks. APoLLo outperforms existing methods by a significant margin and sets a new state-of-the-art on 10 challenging datasets. The page concludes with some future directions for improving the performance and robustness of vision-language models

**Questions For The Authors:**

See weakness

**Reasons To Accept:**

1. The APoLLo achieves state-of-the-art performance on multiple-benchmarks
2. The authors first combine prompt tuning and adaptor on foundational models and nicely design the contrastive loss for intra and inter-modalities with sufficient ablation study to prove its effectiveness.
3. The augmentation method is simple and effective, and more interesting work could base on the authors' work as generative models evolve fast recently.

**Reasons To Reject:**

1. I am wondering why the author chose GPT-2 as their LLM text augmentor while instructBLIP and MiniGPT4 are already available, but this concern is minor
2. Could you explain why not apply the adaptor on the augmented images and texts
3. I am wondering how the cross-attention visualization works on images with multiple objects

**Reproducibility:**

4: Could mostly reproduce the results, but there may be some variation because of sample variance or minor variations in their interpretation of the protocol or method.

**Reviewer Confidence:**

2: Willing to defend my evaluation, but it is fairly likely that I missed some details, didn't understand some central points, or can't be sure about the novelty of the work.

---

> ### Author Rebuttal · Authors · 2023-08-25
>
> We thank the reviewer for the thoughtful and constructive comments. Please find our responses below. The tables here represent mean accuracies reported over all the 10 datasets.
>
> $\textbf{1. Choice of LLM text augmentor}$
>
> GPT-3 has been widely used and validated as a text-augmentor method (Whitehouse et al., 2023, Yin et. al. 2023, Peng et al. 2023,). In particular, Whitehouse et al. (2023) demonstrated the superiority of GPT variants over models like VICUNA (Wei-Lin et al. 2023) as used in InstructBLIP and MiniGPT4. Taking inspiration from these previous works, we chose GPT-3 as the LLM-based text augmentor.
>
> However, as suggested by the reviewer, we also experimented with VICUNA (LLM for InstructBLIP and MiniGPT4) as a text augmentor. Experimental results show that our GPT based (Brown et al. 2020) text augmentation strategy is more superior possibly due to the diversification of the samples generated and the robustness of GPT (as found in Whitehouse et. al. 2023). Table 3 demonstrates the performance comparison when APoLLo is equipped with these two text augmentors:
>
> $\textbf{Table 3. Performance comparison between different LLM based text augmentors}$
>
> | LLM Text Augmentor | Avg. (Base) | Avg. (Novel) |
> | :----: | :----: | :-----: |
> | VICUNA  | 83.16 | 77.29 |
> | **GPT** | **83.77** | **78.03** |
>
>
> $\textbf{2. Ablation on adapter choices}$
>
> Adapter layers were initially added in both augmented image and as well as text branches. However, that increased the number of trainable parameters of the pipeline by nearly two times without substantial performance gains. In fact, our proposed architecture with two adapters (as shown in Fig. 2 in main paper) has performed the best. Please refer to the following table (Table 1 from previous response Reviewer $\textcolor{Blue}{\text{BqGK}}$) for more details:
>
> $\textbf{Table 1. Impact of adding adapters on different image and text branches}$
>
> | Image Branch 1 | Image Branch 2 | Text Branch 1 | Text Branch 2 | Avg. (Base) | Avg. (Novel) |
> | :----: | :----: | :-----: | :--------: | :-----------------: | :-------------: |
> | &check; | - | &check; | - | 83.28 | 77.32 |
> | - | &check; | - | &check; | 83.19 | 77.24 |
> | &check; | - | - | &check; | 83.16 | 77.23 |
> | &check; | &check; | &check; | &check; | 83.77 | 78.01 |
> | **-** | **&check;** | **&check;** | **-** | **83.77** | **78.03** |
>
> $\textbf{3. Cross-attention visualization on images with multiple objects}$
>
> Thank you for bringing this to our notice. As suggested by the reviewers, we provide some examples of the cross-attention visualization on images (from COCO validation set) with multiple objects embedded in the anonymous link (as approved by the program chairs): https://ibb.co/dczvjcN
>
> Here, each of the images contains more than one object category. We generate textual prompts using the same template “A photo of a <CLASS>” for individual objects as shown in the figure. The attention maps are activated in the corresponding regions showing fine grained alignment of APoLLo under challenging scenarios with objects from multiple categories. Please note that here we did a zero-shot transfer on these images with the model trained on ImageNet using our method.
>
> $\textbf{References}$
>
> LLM-powered Data Augmentation for Enhanced Crosslingual Performance (Whitehouse et al. 2023)
>
> TTIDA: Controllable Generative Data Augmentation via Text-to-Text and Text-to-Image Models (Yin et al. 2023)
>
> Check Your Facts and Try Again: Improving Large Language Models with External Knowledge and Automated Feedback (Peng et. al. 2023)
>
> Vicuna: An Open-Source Chatbot Impressing GPT-4 with 90\%* ChatGPT Quality (Wei-Lin et al. 2023)
>
> Language models are few-shot learners (Brown et al. 2020)

---

### Official Review · Reviewer_BqGK · 2023-08-12

**Typos Grammar Style And Presentation Improvements:** N/A
**Soundness:** 4

**Excitement:**

4: Strong: This paper deepens the understanding of some phenomenon or lowers the barriers to an existing research direction.

**Missing References:**

N/A

**Paper Topic And Main Contributions:**

This paper proposes a unified adapter and prompt learning to improve Vision Language Models' zero-shot and few-shot capabilities. Specifically, they introduce trainable cross-attention based adapter layers in both modalities. To better train the model to learn cross-modal alignment, this paper first proposes multi-modal augmentation strategy to create positive and negative pairs for intra-modal and cross-modal contrastive learning. Besides, both intra-modal contrastive consistency and inter-modal similarity maximization are proposed for training the model. Empirical results on multiple benchmarks demonstrate the effectiveness of the proposed approach in generalization in zero-shot and few-shot settings.

**Questions For The Authors:**

N/A

**Reasons To Accept:**

1. This paper proposes a framework that unifies adapter and prompt learning to improve Vision Language Models' performance.
2. Data augmentation approach with LLM and diffusion models are used to enhance contrastive learning performance.
3. Good empirical results on multiple benchmarks, and detailed ablation studies support the claims well.
4. The paper is well-written and easy to follow.

**Reasons To Reject:**

1. Both adapters and prompt tunning have been explored in VLP for some time. More ablations on adapter architecture choice might be discussed to bring more novelty to the approach.

**Reproducibility:**

4: Could mostly reproduce the results, but there may be some variation because of sample variance or minor variations in their interpretation of the protocol or method.

**Reviewer Confidence:**

3: Pretty sure, but there's a chance I missed something. Although I have a good feel for this area in general, I did not carefully check the paper's details, e.g., the math, experimental design, or novelty.

---

> ### Author Rebuttal · Authors · 2023-08-25
>
> We thank the reviewer for the thoughtful and constructive comments. Please find our responses below. The tables here represent mean accuracies reported over all the 10 datasets.
>
> $\textbf{1. Ablations on adapter architecture}$
>
> Please refer to Figure 3 (main paper) where we add ablations with varying number of adapter layers and also Table 4 (main paper) which reports the impact of intra-modal contrastive consistency, adapter tuning and cross attention strategies.
>
> Furthermore, our method contains two image and text branches (where Image Branch 1, Image Branch 2, Text Branch 1, Text Branch 2 are in order from left to right in Figure 2 of main paper and represent the augmented image, original image, original text and augmented text respectively). Here, we add the following table which demonstrates effects of adapter layers on different image and text branch combinations. Please note that all these combinations involve both self- and cross-attentions. We obtain similar accuracies in two cases: (a) Image Branch 2 - Text Branch 1 combination, and (b) when all four branches are taken into account. However, we select the former scenario (a) with Image Branch 2 and Text Branch 1 combination because the number of trainable parameters in this scenario is half than that in the latter. Moreover, Image Branch 2 - Text Branch 1 combination of adapters gives the highest (mean) accuracy values on base and novel classes.
>
> $\textbf{Table 1. Impact of adding adapters on different image and text branches}$
>
> | Image Branch 1 | Image Branch 2 | Text Branch 1 | Text Branch 2 | Avg. (Base) | Avg. (Novel) |
> | :----: | :----: | :-----: | :--------: | :-----------------: | :-------------: |
> | &check; | - | &check; | - | 83.28 | 77.32 |
> | - | &check; | - | &check; | 83.19 | 77.24 |
> | &check; | - | - | &check; | 83.16 | 77.23 |
> | &check; | &check; | &check; | &check; | 83.77 | 78.01 |
> | **-** | **&check;** | **&check;** | **-** | **83.77** | **78.03** |
>
> We add the following ablations w.r.t to the Adapter layers:
> (a) Adapter layers having no attention
> (b) Adapter layers with only self-attention
> (c) Adapter layers with self + cross attention
>
> Experimental results show the optimal performance is achieved in (c) when cross attention is employed between the adapters leading to better multi-modal alignment.
>
>
> $\textbf{Table 2. Impact of incorporating attentions in the adapter layers}$
>
> | No-Attn | Self | Cross | Avg. (Base) | Avg. (Novel) |
> | :----: | :----: | :-----: | :--------: | :-----------------: |
> | &check; | - | - | 82.54 | 76.98 |
> | - | &check; | - |  83.05 | 77.22 |
> | **-** | **&check;** | **&check;** | **83.77** | **78.03** |
>
> We will add these tables in the supplementary of the camera ready version and make the necessary changes. Also, as mentioned in the paper, we will make the codebase public upon acceptance to facilitate reproducibility.

---

### Meta-Review · Area_Chair_cdYu · 2023-09-17

**Recommendation:** 4

**Metareview:**

The overall clarity and significance of the paper are strong as indicated by all reviewers. The proposed method of combining adaptors and prompt tuning is novel, and experimental results are significant.

---

### Decision · Program_Chairs · 2023-10-07

**Decision:**

Accept-Main

**Comment:**

The overall clarity and significance of the paper are strong as indicated by all reviewers. The proposed method of combining adaptors and prompt tuning is novel, and experimental results are significant.